# Towards Verified Robustness under Text Deletion Interventions

**Johannes Welbl**[†‡*]   **Po-Sen Huang**[†]   **Robert Stanforth**[†]   **Sven Gowal**[†]
**Krishnamurthy (Dj) Dvijotham**[†]   **Martin Szummer**[†]   **Pushmeet Kohli**[†]
[†]DeepMind, London, UK   [‡]University College London, UK
{welbl,posenhuang,stanforth,sgowal,dvij,szummer,pushmeet}
@google.com

## Abstract

Neural networks are widely used in Natural Language Processing, yet despite their empirical successes, their behaviour is brittle: they are both over-sensitive to small input changes, and under-sensitive to deletions of large fractions of input text. This paper aims to tackle under-sensitivity in the context of natural language inference by ensuring that models do not become more confident in their predictions as arbitrary subsets of words from the input text are deleted. We develop a novel technique for formal verification of this specification for models based on the popular decomposable attention mechanism by employing the efficient yet effective interval bound propagation (IBP) approach. Using this method we can efficiently prove, given a model, whether a particular sample is free from the under-sensitivity problem. We compare different training methods to address under-sensitivity, and compare metrics to measure it. In our experiments on the SNLI and MNLI datasets, we observe that IBP training leads to a significantly improved verified accuracy. On the SNLI test set, we can verify 18.4% of samples, a substantial improvement over only 2.8% using standard training.

## 1 Introduction

Natural language processing (NLP) widely relies on neural networks, a model class known to be vulnerable to adversarial input perturbations (Szegedy et al., 2013; Kurakin et al., 2016). Adversarial samples typically expose over-sensitivity to semantically invariant text transformations (Belinkov & Bisk, 2017; Ettinger et al., 2017), e.g. character flips (Ebrahimi et al., 2018) or paraphrases (Ribeiro et al., 2018b; Iyyer et al., 2018).

Feng et al. (2018) exposed another type of problematic behaviour: deleting large parts of input text can cause a model's confidence to increase; Figure 1 shows an example. That is, reduced sets of input words can suffice to trigger more confident predictions. Such *under-sensitivity* is problematic: neural models can 'solve' NLP tasks without task-relevant textual comprehension skills, but instead fit spurious cues in the data that suffice to form correct predictions. Models might then achieve strong nominal test accuracy on data of the same (biased) distribution as the training set, by exploiting predictive shortcuts that are not representative of the given NLP task at hand. Consequently, they fail drastically when evaluated on samples without these spurious cues (Jia & Liang, 2017; Poliak et al., 2018; Gururangan et al., 2018; Niven & Kao, 2019).

A major issue with identifying reduced inputs is the combinatorially large space of arbitrary text deletions; this can only be searched exhaustively for short sequences. Prior work has considered heuristics like beam search (Feng et al., 2018) or bandits (Ribeiro et al., 2018a), but these are generally not guaranteed to find the worst-case reductions.

In this work, we address the under-sensitivity issue by designing and formally verifying the under-sensitivity specification that a model should not become *more* confident as arbitrary subsets of input words are deleted.[1] Under-sensitivity behaviour is not reflected in nominal accuracy, but one can

---

[*]Work done during an internship at DeepMind.
[1] This specification is discussed in Section 3. Although a conservative choice, we find it is rarely satisfied.

| Original Sample | **Premise:** A little boy in a blue shirt holding a toy.
**Hypothesis:** A boy dressed in blue holds a toy.
Entailment (86.4%) |
|---|---|
| Reduced Sample | **Premise:** A little boy in a blue shirt holding a toy.
**Hypothesis:** A boy dressed in blue holds a toy.
Entailment (91.9%) |

Figure 1: Example of under-sensitive behaviour in Natural Language Inference, where deleting premise words increases model confidence. This problem was identified by Feng et al. (2018); our aim is to formally verify whether or not any such reductions exist, over the combinatorially large space of possibilities.

instead use this specification to measure and evaluate the extent with which samples exhibit under-sensitivity. Instead of better, yet still imperfect search heuristics, we describe how interval bound propagation (IBP) (Gowal et al., 2018; Mirman et al., 2018) – a formal model verification method – can be used to efficiently cover the full reduction space, and verify the under-sensitivity specification. IBP can be applied at test time to arbitrary model inputs to verify whether or not they are under-sensitive; but it can also be used to derive a new auxiliary training objective that leads to models verifiably adhering to this specification, and which we find generalises to held-out test data.

While under-sensitivity has been demonstrated for several NLP tasks (Feng et al., 2018), we chose to study the use case of natural language inference (NLI) (Dagan et al., 2006; Bowman et al., 2015) in particular as a representative task: sequences are comparatively short, datasets large, and the label complexity is small. We investigate the verification of the popular decomposable attention model (DAM)[2] (Parikh et al., 2016) in detail. This architecture covers many of the neural layer types of contemporary models, and we focus on a detailed description for how IBP can be leveraged to efficiently verify its behaviour. We then experimentally compare various training methods addressing under-sensitivity: i) standard training ii) data augmentation iii) adversarial training iv) IBP-verified training and v) entropy regularisation, and evaluate their effectiveness against nominal (test) accuracy, adversarial accuracy, IBP-verified accuracy and a verification oracle.

To summarise, the main contributions of this paper are (1) Formalisation of the problem of verifying an under-sensitivity specification, (2) Verification of the Decomposable Attention Model using Interval Bound Propagation, and (3) Empirical analysis of the efficacy of (i) different evaluation methods for verifying robustness; and (ii) different training methods for developing verifiably robust models.

## 2 RELATED WORK

**Natural Language Inference.** Natural Language Inference (Dagan et al., 2006) is the task of predicting whether a natural language premise entails a natural language hypothesis. The availibility of large-scale datasets (Bowman et al., 2015; Williams et al., 2018) has spurred a profusion of neural architecture development for this task, e.g. (Rocktäschel et al., 2016; Parikh et al., 2016; Chen et al., 2017), among many others.

**Adversarial Vulnerability in NLP.** There is a growing body of research into NLP adversarial examples, each using a slightly different choice of semantically invariant text transformations, or a task-specific attack. A first class of attack considers word- and character-level perturbation attacks (Ebrahimi et al., 2018; Alzantot et al., 2018) while another type of attack exploits back-translation systems to either mine rules (Ribeiro et al., 2018b) or train syntactically controlled paraphrasing models (Iyyer et al., 2018). Li et al. (2017) use syntactic and lexical transformations, whereas Belinkov & Bisk (2017) investigate synthetic and natural noise in Machine Translation. Jia & Liang (2017) and Mudrakarta et al. (2018) introduce task-specific adversarial attacks for Reading Comprehension/QA, Zhao et al. (2018) for Machine Translation and NLI, and Thorne & Vlachos (2019) for Fact Checking. In NLI in particular, Minervini & Riedel (2018) penalise adversarially chosen

---

[2]The DAM model is $\approx 5\%$ behind the current state-of-the-art on SNLI. We see specification verification for this model as a stepping stone towards verifying larger attention-based NLP architectures, such as BERT.

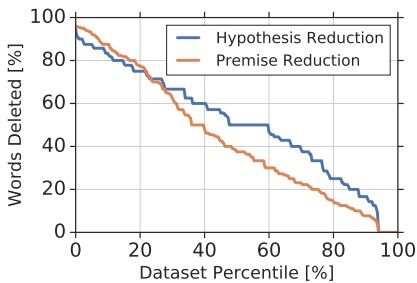

Figure 2: Under-sensitivity: This figure maps dataset percentiles against the proportion of words one can delete before DAM prediction confidence decreases, where reduced samples are found using beam search.

logical inconsistencies in NLI predictions, Kang et al. (2018) use background-knowledge guided adversaries, and Glockner et al. (2018) utilise lexical entailment relationships.

Ribeiro et al. (2016) and Ribeiro et al. (2018a) describe analysis tools that have uncovered model over-sensitivity and under-sensitivity, respectively. Feng et al. (2018) focus on under-sensitivity, showing that models can become more confident as large fractions of input text are deleted, whereas Niu & Bansal (2018) address under-sensitivity in a dialogue setting. Jacobsen et al. (2019) demonstrated a link between excessive prediction invariance and model vulnerability in computer vision.

**Formal Verification.** Formal verification provides a provable guarantee that models are consistent with a formally defined *specification* (a mathematical relationship between the inputs and outputs of the model). Examples of specifications include robustness to bounded adversarial perturbations, monotonicity of the output with respect to a subset of the inputs, and consistency with physical laws (Qin et al., 2019). Literature can be categorised into complete methods that use Mixed-Integer Programming (MIP) (Bunel et al., 2017; Cheng et al., 2017) or Satisfiability Modulo Theory (SMT) (Katz et al., 2017), and incomplete methods that solve a convex relaxation of the verification problem (Weng et al., 2018; Wong & Kolter, 2018; Wang et al., 2018; Raghunathan et al., 2018b). Complete methods perform exhaustive enumeration to find a counter-example to the specification or rule out the existence of counter-examples (thus proving that the specification is true). Hence, complete methods are expensive and difficult to scale. Incomplete methods are conservative (i.e. they cannot always prove that a specification is true even when it is), but are more scalable and can be used inside the training loop for training models to be consistent and verifiable (Raghunathan et al., 2018a; Wong & Kolter, 2018; Dvijotham et al., 2018a; Gowal et al., 2018; Dvijotham et al., 2018b).

While Barr & Klavans (2001) address the issue of verification in NLP, most of the recent work has focused on $\ell_\infty$ norm-bounded perturbations for image classification. This paper complements work on incomplete verification methods by extending IBP to NLI where inputs are inherently discrete (to the contrary of images which are continuous). In the NLP context in particular, Huang et al. (2019) and Jia et al. (2019) have very recently verified CNN, and LSTM models with specifications against over-sensitivity adversaries under synonym replacement. Wang et al. (2019) study verification of output length specifications in machine translation models, showing that the outputs of machine translation and image captioning systems can be provably bounded when the inputs are perturbed within a given set. In contrast, this work examines under-sensitivity behaviour: excessive model prediction invariance under arbitrary word combination deletions. We highlight that the verification of neural networks is an extremely challenging task, and that scaling complete and incomplete methods to large models is an open problem.

## 3    FORMULATING A SPECIFICATION AGAINST UNDER-SENSITIVITY

Neural networks are expressive models that can fit large datasets and achieve strong nominal test accuracy. At the same time however, they can fit data in a way that violates our idea of how they should fit it, from an input attribution perspective. Figure 2 visualises the extent of the problem in NLI: it is, for example, for 20% of the SNLI test set possible to delete 78% or more of premise

words while the prediction confidence increases or remains the same. We will next formally describe a specification that checks a model's under-sensitivity, i.e. whether any such reduction exists.

The specification addresses model output probabilities when parts of the input text are deleted. To this end, we first introduce the notion of a perturbation space $\mathcal{X}^{\mathrm{in}}(\mathbf{x}^{\mathrm{nom}})$ of an original *nominal* input $\mathbf{x}^{\mathrm{nom}}$. This space contains all possible reductions, i.e. inputs where arbitrarily chosen tokens of the original nominal input $\mathbf{x}^{\mathrm{nom}}$ are deleted. Note that this space grows exponentially in the length of the input. We would like to verify whether or not there exists any reduced input $\mathbf{x}^-$ with higher probability for the (nominal) prediction than $\mathbf{x}^{\mathrm{nom}}$ has. More formally, this can be stated as a specification:

$$\forall \mathbf{x}^- \in \mathcal{X}^{\mathrm{in}}(\mathbf{x}^{\mathrm{nom}}) : P(\hat{y}|\mathbf{x}^-) \leq P(\hat{y}|\mathbf{x}^{\mathrm{nom}}) \tag{1}$$

where $\hat{y}$ is the (nominal) model prediction.

Determining precisely how prediction probabilities should change when input words are deleted is contentious and prone to inconsistencies: removing stop words for example may lead to little relevant change, while crucial information carrying words (e.g., '*not*') can significantly alter the meaning of the sentence in a task. It is important to be cautious and not too restrictive in the specification design, and certain that whatever is specified is desirable. A specification to at least *not increase* prediction probabilities under arbitrary input deletion is a conservative choice.[3] Other specifications are worth consideration, such as monotonically decreasing certainty as more input is deleted. We will however see that even our very conservative choice of an under-sensitivity specification is hard to positively verify for most inputs in the DAM model.

There are different approaches to establish if the Specification (1) is satisfied. With unlimited computational capacity, the property could exhaustively be evaluated for all $\mathbf{x}^- \in \mathcal{X}^{\mathrm{in}}(\mathbf{x}^{\mathrm{nom}})$. Statistically sampling from the reduction space can give an indication of under-sensitivity, but has a very limited coverage rate. Search heuristics can try to identify violations (and be used for 'adversarial' training), but there is no guarantee that a stronger search procedure cannot find more or worse violations. IBP verification on the other hand offers a formal guarantee across the whole space by establishing outer bounds for $\mathcal{X}^{\mathrm{in}}(\mathbf{x}^{\mathrm{nom}})$ and resulting bounds on output probabilities.

## 4 BACKGROUND

We next give a brief introduction to the Decomposable Attention Model (DAM) which we will later verify. The DAM architecture comprises commonly used neural NLP components, such as word embeddings, attention, and feed-forward networks. Subsequently we introduce Interval Bound Propagation and then bring these together to verify the behaviour of the DAM, i.e. efficiently assert whether an input satisfies Specification (1).

**Decomposable Attention.** The NLI task takes two word sequences as input – a premise and a hypothesis – and outputs a discrete entailment label prediction in {*entailment*, *neutral*, *contradiction*}. The DAM architecture (Parikh et al., 2016) assumes the input word sequences to be embedded (e.g. as $d$-dimensional word vectors), i.e. operates on two sequences of input vectors:[4] $\mathbf{A} = [\mathbf{a}_1; \ldots; \mathbf{a}_I] \in \mathbb{R}^{d \times I}$, and $\mathbf{B} = [\mathbf{b}_1; \ldots; \mathbf{b}_J] \in \mathbb{R}^{d \times J}$, where $[.;.]$ denotes concatenation, and $I$ and $J$ are sequence lengths. Word vectors are individually transformed with a vector-valued function $F(.)$, and pairs thereof are then compared:

$$e_{ij} = F(\mathbf{a}_i)^\top F(\mathbf{b}_j) \in \mathbb{R} \tag{2}$$

Note that we follow the notation of Parikh et al. (2016), and that $e_{ij}$ is not related to a basis vector. In the general model formulation $F$ can be a linear transformation or MLP; this does not affect the derivations made here. Adopting matrix notation across position pairs $(i, j)$, Equation (2) can instead be rewritten in matrix form as $\mathbf{E} = F(\mathbf{A})^\top F(\mathbf{B}) \in \mathbb{R}^{I \times J}$, which is used to compute two

---

[3]We do note that removing relativising words (e.g. '*somewhat*') can present an exception to this specification, but they are very rare in the datasets used here.

[4]We will use boldface notation for vectors and matrices, and light notation for scalars and functions.

attention masks – one over each sequence – by normalising across $i$ or across $j$:

$$P_{ij}^{(\mathbf{A})} = \frac{\exp(e_{ij})}{\sum_k \exp(e_{kj})}; \quad \mathbf{P}^{(\mathbf{A})} \in \mathbb{R}^{I \times J} \tag{3}$$

$$P_{ij}^{(\mathbf{B})} = \frac{\exp(e_{ij})}{\sum_k \exp(e_{ik})}; \quad \mathbf{P}^{(\mathbf{B})} \in \mathbb{R}^{I \times J} \tag{4}$$

These two attention masks serve as coefficients in a convex combination over the original word vectors, aggregating each of the two sequences:

$$\mathcal{A} = \mathbf{A} \cdot \mathbf{P}^{(\mathbf{A})} \in \mathbb{R}^{d \times J}$$
$$\mathcal{B} = \mathbf{B} \cdot (\mathbf{P}^{(\mathbf{B})})^\top \in \mathbb{R}^{d \times I}$$

That is, $\mathcal{A}$ and $\mathcal{B}$ hold attention-aggregated word vectors from $\mathbf{A}$ and $\mathbf{B}$ at positions $j = 1, \ldots, J$ and $i = 1, \ldots, I$, respectively. These are joined with the original word representations, mixed using a position-wise feed-forward network $G : \mathbb{R}^{2d} \to \mathbb{R}^{d'}$ and finally summed into a single vector representation for each sequence:

$$\mathbf{v}_1 = \sum_i G([\mathbf{a}_i; \mathcal{B}_i]) \in \mathbb{R}^{d'} \tag{5}$$

$$\mathbf{v}_2 = \sum_j G([\mathcal{A}_j; \mathbf{b}_j]) \in \mathbb{R}^{d'} \tag{6}$$

As a last step, a logit vector with entries for each class is computed as $H([\mathbf{v}_1, \mathbf{v}_2])$, where $H : \mathbb{R}^{2d'} \to \mathbb{R}^C$ is again a feed-forward network, and $C$ is the number of output classes.

**Interval Bound Propagation.** IBP is an incomplete but efficient verification method that can be used to verify input-output relationships. It tracks how a part of the input space (in our case: the perturbation space $\mathcal{X}^{\text{in}}(\mathbf{x}^{\text{nom}})$) propagates forward through the network. IBP starts with an axis-aligned bounding box surrounding $\mathcal{X}^{\text{in}}(\mathbf{x}^{\text{nom}})$, and uses interval arithmetic to obtain an axis-aligned bounding box for the output set. Formally, let us assume that the neural network is defined by a sequence of transformations $h_k$ for each of its $K$ layers. That is, for $\mathbf{z}_0 \in \mathcal{X}^{\text{in}}(\mathbf{x}^{\text{nom}})$

$$\mathbf{z}_k = h_k(\mathbf{z}_{k-1}) \quad k = 1, \ldots, K \tag{7}$$

The output $\mathbf{z}_K \in \mathbb{R}^C$ has $C$ logits corresponding to $C$ classes. IBP bounds the activation $\mathbf{z}_k$ of each layer by an axis-aligned bounding box (i.e., $\underline{\mathbf{z}}_k \leq \mathbf{z}_k \leq \overline{\mathbf{z}}_k$) using interval arithmetic. We have for each coordinate $z_{k,i}$ of $\mathbf{z}_k$:

$$\underline{\mathbf{z}}_{k,i} = \min_{\underline{\mathbf{z}}_{k-1} \leq \mathbf{z}_{k-1} \leq \overline{\mathbf{z}}_{k-1}} \mathbf{e}_i^\top h_k(\mathbf{z}_{k-1})$$
$$\overline{\mathbf{z}}_{k,i} = \max_{\underline{\mathbf{z}}_{k-1} \leq \mathbf{z}_{k-1} \leq \overline{\mathbf{z}}_{k-1}} \mathbf{e}_i^\top h_k(\mathbf{z}_{k-1}) \tag{8}$$

where $\mathbf{e}_i$ is the standard $i^{\text{th}}$ basis vector. Finally, at the last layer, an upper bound on the worst-case violation of the specification can be evaluated quickly from the logit lower and upper bounds $\underline{\mathbf{z}}_K$ and $\overline{\mathbf{z}}_K$ respectively, as the bounds translate directly into bounds for the softmax probabilities.

IBP can be performed in parallel while running the nominal forward pass. However in general the output bounds are loose, which is exacerbated with increasing network depth. Consequently IBP over-approximates the true extent of the image of $\mathcal{X}^{\text{in}}(\mathbf{x}^{\text{nom}})$ in output space, and can result in false negatives. It is thus in practice important to keep the bounds as tight as possible. IBP can be used at test time for verification, but also for training, minimising loss terms derived from the logit bounds. IBP has been used on MLPs and convolutional networks with monotonic activations (Gowal et al., 2018; Huang et al., 2019). One technical contribution of this work is to apply it to a model with an attention component (Section 5).

## 5 VERIFYING UNDER-SENSITIVITY GUARANTEES FOR THE DAM MODEL

To address under-sensitivity, we aim to verify Specification (1) for the DAM model. If the upper probability bound $\overline{\mathbf{z}}_K$ of the entire perturbation space $\mathcal{X}^{\text{in}}(\mathbf{x}^{\text{nom}})$ is smaller than the probability $P(\hat{y}|\mathbf{x}^{\text{nom}})$ for the predicted class, then the specification is verified. That is $\forall \mathbf{z}_0 \in \mathcal{X}^{\text{in}}(\mathbf{x}^{\text{nom}})$:

$$P(\hat{y}|\mathbf{z}_0) \leq \overline{\mathbf{z}}_{K,\hat{y}} \leq P(\hat{y}|\mathbf{x}^{\text{nom}}) \tag{9}$$

Using this inequality, we can assert whether Specification (1) is verifiably satisfied for any given $\mathbf{x}^{\text{nom}}$, i.e. whether there exist any reduced samples with higher probability than $\mathbf{x}^{\text{nom}}$.

**Overview.** We will first describe the model behaviour when removing a single word at fixed position, and then extend this to deleting single words at any position, and finally generalise this to arbitrary multi-token deletions.

One key difference to IBP bounds of other architectural components, such as CNNs or feed-forward layers, is the need for bounds on the attention normalisation, which has to take into account per-token upper and lower bounds. We will exploit the fact that each vector of $\mathcal{B}$ is a convex combination of the $J$ vectors that constitute $\mathbf{B}$ (and similarly for $\mathcal{A}$). Hence, component-wise bounds on $\mathcal{B}$ can be obtained efficiently by maximising over those $J$ vectors. $\mathcal{A}$ and $\mathcal{B}$ are then inputs to a regular feed-forward network ($G$ followed by $H$), for which IBP can be used.

**Deleting Single Word: Particular Position.** We first describe how model variables behave when an individual token at a fixed position $r$ is removed from one of the sequences. Without loss of generality, we delete words from the second sequence, noting that the model has a symmetric architecture and that the same can be derived for the other input sequence. We denote all resulting quantities with a bar (as in $\bar{\mathbf{B}}$). That is, when removing a single token at position $r$:

$$\bar{\mathbf{B}} = [\mathbf{b}_1, \ldots, \mathbf{b}_{r-1}, \mathbf{b}_{r+1}, \ldots, \mathbf{b}_J] \in \mathbb{R}^{d \times (J-1)} \tag{10}$$

whereas $\bar{\mathbf{A}} = \mathbf{A}$. Since $F(.)$ is applied per-position in the sequence, the effect of word deletion remains isolated at this point; the matrix product $\bar{\mathbf{E}} = F(\bar{\mathbf{A}})^\top F(\bar{\mathbf{B}}) \in \mathbb{R}^{I \times (J-1)}$ has identical entries as before, but the $r$-th column disappears.

**Renormalising Attention Masks.** Likewise the attention mask $\bar{\mathbf{P}}^{(\mathbf{A})}$ has identical entries compared to $\mathbf{P}^{(\mathbf{A})}$, but the $r$-th column removed. That is, for $i = 1, \ldots, I$ and $j = 1, \ldots, J$ s.t. $j \neq r$:

$$\bar{P}_{ij}^{(\mathbf{A})} = \frac{\exp\left(\bar{e}_{ij}\right)}{\sum_k \exp\left(\bar{e}_{kj}\right)}; \quad \bar{\mathbf{P}}^{(\mathbf{A})} \in \mathbb{R}^{I \times (J-1)} \tag{11}$$

The attention mask $\bar{\mathbf{P}}^{(B)}$ on the other hand has renormalised entries. The values retain their relative order, yet the entries are larger because the $r$-th normalisation summand is removed. For $j \neq r$:

$$\bar{P}_{ij}^{(\mathbf{B})} = \frac{\exp\left(e_{ij}\right)}{\sum_{k \neq r} \exp\left(e_{ik}\right)}; \quad \bar{\mathbf{P}}^{(\mathbf{B})} \in \mathbb{R}^{I \times (J-1)} \tag{12}$$

Hence we can compute $\bar{P}_{ij}^{(\mathbf{B})}$ in closed form as

$$\bar{P}_{ij}^{(\mathbf{B})} = P_{ij}^{(\mathbf{B})} \cdot \frac{\sum_k \exp\left(e_{ik}\right)}{\sum_{k \neq r} \exp\left(e_{ik}\right)} \tag{13}$$

To summarise the above: attention weights $P_{ij}^{(\mathbf{B})}$ remain largely unchanged when deleting token $r$, but are rescaled to take into account missing normalisation mass.

In the next step, the model computes convex combinations $\bar{\mathcal{A}}$ and $\bar{\mathcal{B}}$. Concretely, $\bar{\mathcal{A}} = \bar{\mathbf{A}} \cdot \bar{\mathbf{P}}^{(\mathbf{A})} \in \mathbb{R}^{d \times (J-1)}$ has unchanged elements compared to before (as $\mathbf{A}$ remains unchanged), but the $r$-th column is removed. For $\bar{\mathcal{B}} = \bar{\mathbf{B}} \cdot (\bar{\mathbf{P}}^{(\mathbf{B})})^\top \in \mathbb{R}^{d \times I}$ the dimensionality remains unchanged, but $\bar{\mathbf{B}}$ has fewer elements and $(\bar{\mathbf{P}}^{(\mathbf{B})})^\top$ is renormalised accordingly. Note that all this can still be computed in closed form using Equation (13), i.e. without need for IBP thus far, and these quantities can further be fed through the remaining network layers $G$ and $H$ to obtain concrete probabilities.

**Deleting Single Word: Arbitrary Position.** We have reached the point where $\bar{\mathbf{A}}, \bar{\mathbf{B}}, \bar{\mathcal{A}}$ and $\bar{\mathcal{B}}$ are derived in closed form, for fixed position $r$. These can be computed exactly without approximation, for deleted words at any position $r$ in the sequence. Extending this to arbitrary single-word deletions, we take the elementwise minimum / maximum across all possible single word deletions, e.g.

$$\bar{\mathcal{B}}^{\text{Upper}} = \max_{r=1,\ldots,J} \bar{\mathcal{B}}(r) \tag{14}$$

$$\bar{\mathcal{B}}^{\text{Lower}} = \min_{r=1,\ldots,J} \bar{\mathcal{B}}(r) \tag{15}$$

which establishes upper and lower bounds for each element, and analogously for the other matrices.

In the DAM architecture, these matrices are next fed into dense feed-forward layers $G$ (Equations (5) and (6)) and $H$, each with two layers.[5] We use IBP to propagate bounds through these layers, feeding in bounds on $\bar{\mathbf{A}}$, $\bar{\mathbf{B}}$, $\bar{\mathcal{A}}$ and $\bar{\mathcal{B}}$ as described above. As a result, after propagating these bounds through $G$ and $H$, we obtain bounds on output logits (and consequently on probabilities) for deletions of a single token at any position.

One further simplification is possible: we compute $\bar{\mathbf{v}}_2$ directly from $\mathbf{v}_2$ by subtracting the $r$-th summand for fixed $r$ (see Equation (6)). Generalising this to arbitrary positions $r$, we can bound the subtracted vector with $\max_{r=1,\dots,J}\{G([\bar{\mathcal{A}}_r; \bar{\mathbf{b}}_r])\}$ and $\min_{r=1,\dots,J}\{G([\bar{\mathcal{A}}_r; \bar{\mathbf{b}}_r])\}$, and thus directly obtain bounds for $\bar{\mathbf{v}}_2$.

**Deleting Several Words.** We have described the behaviour of intermediate representations (and bounds for them) under deletions of arbitrary individual words; the case of removing several words is similar. The values of remaining individual word vectors $\mathbf{a}_i$ and $\mathbf{b}_j$ naturally remain unchanged. The previously established bounds for single word deletions can be partly re-used to establish bounds for arbitrary multi-word deletions, see appendix A for more detail. The resulting bounds for $\bar{\mathbf{v}}_1$ and $\bar{\mathbf{v}}_2$ are then input to a regular feed-forward network, for which IBP can be used.

## 6 EXPERIMENTS

We now evaluate to which extent the DAM model verifiably satisfies the Specification (1) against under-sensitivity, and we will furthermore compare different training approaches. Experiments are conducted on two large-scale NLI datasets: SNLI (Bowman et al., 2015) and multiNLI (Williams et al., 2018), henceforth MNLI. Whereas Feng et al. (2018) addressed deletions of hypothesis words in SNLI, we establish the phenomenon also for MNLI, and for premise reductions. In our experiments we use premise reductions, noting that under-sensitivity is also present for hypotheses (see Fig. 2). For SNLI we use standard dataset splits, tuning hyperparameters on the development set and reporting results for the test set. For MNLI we split off 2000 samples from the development set for validation purposes and use the remaining samples as test set. We use the same types of feed-forward components, layer size, dropout, and word embedding hyperparameters described by Parikh et al. (2016).

**Evaluation Metrics** We evaluate with respect to the following metrics:

1. *Accuracy:* Standard test accuracy.
2. *Verified Accuracy:* This metric measures whether both i) the prediction is correct ii) it can be verified that no reduction with higher probability exists, using IBP verification.
3. *Beam Search Heuristic:* This metric uses beam search to find specification violations in the perturbation space, following the protocol of Feng et al. (2018). Search begins from the full sequence, gradually deleting words while keeping a beam of width 10. This metric then measures whether both i) the search heuristic found *no* counterexample, and ii) the prediction is correct. Note that this heuristic does not cover the full perturbation space, i.e. does not suffice to rule out counterexamples to the specification. This metric provides an upper bound for verified accuracy.

**Training Methods** We will compare the following training methods:

1. *Standard Training:* This provides a baseline for under-sensitivity behaviour under standard log-likelihood training.
2. *Data Augmentation:* A first and comparatively simple way to address under-sensitivity is by adding training samples with random word subsets deleted, and penalising the model with a loss proportional to the specification violation.
3. *Adversarial Training:* Here we use a more systematic approach than random word deletions: we search within the perturbation space for inputs with large differences between

---

[5]We use *softplus* activations, a continuous approximation to the ReLU used in the original model.

| Training Method | Accuracy | Verified Accuracy | Beam Search Heuristic |
|---|---|---|---|
| Standard Training | 77.22 | 2.83 | 3.36 |
| Data Augmentation | 76.37 | 5.09 | 6.27 |
| Adversarial Training: random | 76.89 | 1.79 | 4.16 |
| Adversarial Training: beam search | 76.09 | 5.48 | **23.76** |
| Entropy Regularisation | **77.32** | 5.82 | 6.28 |
| IBP-Training | 75.51 | **18.36** | 19.26 |

(a) SNLI

| Training Method | Accuracy | Verified Accuracy | Beam Search Heuristic |
|---|---|---|---|
| Standard Training | 60.00 | 7.77 | 8.77 |
| Data Augmentation | **62.02** | 1.93 | 4.26 |
| Adversarial Training: random | 61.89 | 2.60 | 5.04 |
| Adversarial Training: beam search | 58.74 | 0.45 | 7.44 |
| Entropy Regularisation | 60.74 | 8.83 | 9.47 |
| IBP-Training | 44.95 | **17.44** | **19.07** |

(b) MNLI

Table 1: Experimental results: accuracy vs. verified accuracy using IBP, for different training methods. All models tuned for verified accuracy, numbers in %.

nominal prediction probability and reduced probability, i.e. the strongest specification violations. We compare both i) random adversarial search that samples 512 randomly reduced perturbations and picks the strongest violation, and ii) beam search with width 10, following the protocol of Feng et al. (2018). Both for data augmentation and adversarial training, altered samples are recomputed throughout training.

4. *Entropy Regularisation:* Feng et al. (2018) observed that entropy regularisation on prediction probabilities can partially mitigate the severity of under-sensitivity.

5. *IBP-Training:* Here we use IBP verification as described in Section 5, which provides upper bounds on the prediction probability of arbitrarily reduced inputs (Eq. (9)). We penalise the model with an auxiliary hinge loss on the difference between the upper probability bound for the gold label $y$ and the nominal probability $P(y|\mathbf{x}^{\text{nom}})$. Note that the upper bound serves as a proxy for the adversarial objective, as it over-approximates the probabilities of arbitrary reduced samples, covering the full reduction space comprehensively.

**Training Details** The training methods described above make use of an additive contribution to the training loss besides standard log-likelihood. We tune the scale of the respective contribution in [0.01, 0.1, 1.0, 10.0, 100.0]. All experiments used a learning rate of 0.001, Adam optimiser, and batch size 128. We perform early stopping with respect to verified accuracy, for a maximum of 3M training steps. For verified training, we found it useful to continuously phase in the volume of the perturbation space to its maximum size, similar to Gowal et al. (2018). Concretely, we compute the per-dimension center of upper and lower bound, and start linearly increasing its volume until it reaches the full perturbation space volume. Similarly we phase in the perturbation radius, i.e. the maximum number of words deleted from 1 to the maximum sequence length of 48. We tune phase-in intervals in $[10^0, 10^3, 10^4, 10^5, 10^6]$ training steps. We also experimented with over-inflating the perturbation volume to larger than its real size at training time, as well as randomly sampling a maximum perturbation radius during training, neither of which improved verifiability results.

## 6.1 RESULTS AND ANALYSIS

**Evaluating the Effectiveness of IBP for Verification.** Tables 1a and 1b show the main results. For both datasets, a non-negligible portion of data points can be verified using IBP. The gap between (standard) accuracy however is striking: only a small fraction of correctly predicted inputs is actually verifiably not under-sensitive. Note that IBP accuracy is naturally bounded above by the beam search heuristic, which does however not cover the full reduction space, and overestimates verification

| Metric | Time[s] | # Eval's /sample |
|---|---|---|
| Accuracy | 2 | 1 |
| IBP Verification | 3 | $\approx 2$ |
| Verification Oracle | – | $2^L$ |
| Oracle up to 200K | 45674 | 200000 |
| Beam Search | 505 | $\approx b \cdot L$ |

Table 2: Computational cost of verification. Left: time elapsed (1 GPU) for evaluating 300 SNLI samples, without cross-sample batching. Right: worst-case number of forward passes. $L$: sequence length; $b$: beam width.

| Training | IBP | Oracle | Beam |
|---|---|---|---|
| Standard Training | 4.34 | 5.13 | 5.37 |
| Data Augmentation | 6.48 | 8.59 | 8.78 |
| Adversarial:random | 1.87 | 6.88 | 7.03 |
| Adversarial:beam | 5.13 | **31.90** | **32.14** |
| Entropy Regul. | 8.35 | 8.90 | 9.28 |
| IBP-Training | **19.29** | 20.68 | 20.94 |

Table 3: Oracle on SNLI sequences up to 12 tokens. Numbers in %.

rates. IBP verification becomes particularly effective when adding the IBP-verifiability objective during training, verifying 18.36% and 17.44% of samples on SNLI and MNLI. Verifiability does however come at a cost: test accuracy is generally decreased when tuning for verifiability, compared to Parikh et al. (2016). This highlights a shortcoming of test accuracy as a metric: it does not reflect the under-sensitivity problem. Once under-sensitivity is taken into account by dedicated training objectives or tuning for verification rates, nominal accuracy suffers.

**Computational Efficiency of IBP Verification.** Table 2 gives a breakdown of the computational cost incurred for verification, both empirically, and the theoretical worst-case number of forward passes required per sample. IBP verification comes with small computational overhead compared to a standard forward pass, which is incurred by propagating upper and lower interval bounds through the network once. A full oracle is computationally infeasible, instead we used an exhaustive search oracle, but only up to a maximum budget of 200K forward passes per sample. Even when stopping as soon as a single reduced sample is found that violates the specification, the incurred time is orders of magnitude larger than verification via IBP.

**Comparing Training Methods.** We next discuss the differences between training methods, and how they reflect in verified model behaviour. In absolute terms, standard training does not adhere to the under-sensitivity specification well, neither on SNLI nor MNLI. Data augmentation and random adversarial training lead to slightly different results on the two datasets, albeit without major improvements. These methods have a strong random component in their choice of deletions, and this tends to lead to lower verification rates on MNLI, where premises are on average 6.2 tokens longer, and the reduction space is correspondingly larger. Beam search adversarial training leads to improved verification rates on SNLI, yet not for MNLI, and it is noteworthy that when also trained with beam search adversarial samples, beam search evaluation improves substantially. Entropy regularisation improves verified accuracy over standard training; this is in line with previous observations that it mitigates under-sensitivity behaviour, made by Feng et al. (2018). Finally, the dedicated IBP-Training objective substantially raises verification rates compared to all other approaches. In an ablation (Table 3) we evaluate performance on short sequences (up to 12 tokens) in the SNLI test set: here an exhaustive search over all possible reductions is feasible. Still the absolute verification rates are low in absolute terms, but we observe that shorter sequences are comparatively easier to verify, and that the incomplete IBP verification can approach the verification levels of the complete oracle (see rows 1,2,5, and 6). For adversarial training (rows 3 and 4), however, oracle verification rates are much closer to the Beam Search Heuristic. This suggests that i) for short sequences the smaller perturbation space can be covered better by beam search, and ii) adversarial training can

lead to high verifiability on short sequences, but it fits a model in a way that results in loose IBP bounds.

## 7 DISCUSSION

Verification of a specification offers a stronger form of robustness than robustness to adversarial samples. Adversarial accuracy, as e.g. derived from beam search, might conceptually be easier to compute, yet has no guarantees to find all or the strongest violations. In fact, evaluating against weak adversaries under-estimates the extent of a problem (Uesato et al., 2018) and may lead to a false sense of confidence. IBP verification can provide guarantees on the nonexistence of reduced inputs, but it is incomplete and can have false negatives.

Observations of comparatively low verification or adversarial accuracy rates – as in this work – are not new, and have been found to be a general problem of datasets with high sample complexity (Schmidt et al., 2018). We emphasise that under-sensitivity is a very challenging problem to address; even the relatively conservative specification of non-increasing probability under deletion cannot be fulfilled for the majority of test samples under the baselines tested.

We see the verification of the attention-based DAM model as a stepping stone towards the verification of larger and more performant attention-based architectures, such as BERT. Following the derivations here, token deletion bounds could similarly be propagated through BERT's self-attention layer. Towards this end, however, we see two main hurdles: i) BERT's network depth, resulting in gradually looser IBP bounds ii) BERT's word piece tokenisation, which requires special consideration in conjunction with token-level perturbations.

## 8 CONCLUSION

We have investigated under-sensitivity to input text deletions in NLI and recast the problem as one of formally verifying a specification on model behaviour. We have described how Interval Bound Propagation can be used in order to verify the popular Decomposable Attention Model, and have then compared several training methods in their ability to address and be verified against under-sensitivity. We observed that only a relatively small fraction of data points can be positively verified, but that IBP-training in particular is capable of improving verified accuracy.

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

# A APPENDIX: MULTI-WORD DELETION BOUNDS

In this section we will elaborate on how bounds for $\bar{\mathbf{v}}_1$ and $\bar{\mathbf{v}}_2$ are computed in the case of arbitrary multi-word deletions.

**Bounds on $\bar{\mathbf{v}}_1$:**  Recall from Equation 5 that

$$\mathbf{v}_1 = \sum_i G([\mathbf{a}_i; \mathcal{B}_i]) \in \mathbb{R}^{d'}$$

We can compute bounds on $\bar{\mathbf{v}}_1$ under arbitrary multi-word deletions by bounding both $\bar{\mathbf{a}}_i$ and $\bar{\mathcal{B}}_i$, and then propagating these bounds using IBP. The values of $\mathbf{a}_i$ remain unchanged under deletions in the second input sequence ($\bar{\mathbf{a}}_i = \mathbf{a}_i$), and we will thus focus on deriving upper and lower bounds for $\bar{\mathcal{B}}_i$.

To this end, recall that individual columns in $\mathcal{B}$ are computed as convex combination of vectors in $\mathbf{B}$. That is, the $i^{th}$ column is computed as $\mathcal{B}_i = \sum_j \mathbf{b}_j P_{i,j}^{(B)}$, where $P_{i,j}^{(B)}$ is the entry of $\mathbf{P}^{(\mathbf{B})}$ at position $(i,j)$. The (elementwise) minimum and maximum values that $\bar{\mathcal{B}}_i$ can assume, are given by the (elementwise) minimum and maximum of single vectors $\mathbf{b}_j$: $\mathbf{b}_{min} = \min_j\{\mathbf{b}_j\}$, and $\mathbf{b}_{max} = \max_j\{\mathbf{b}_j\}$. The values of $\bar{\mathcal{B}}_i$ are hence bounded elementwisely from above by the values in $\mathbf{b}_{max}$:

$$\bar{\mathcal{B}}_i = \sum_{j \notin D} \mathbf{b}_j \bar{P}_{i,j}^{(\bar{B})} \leq \sum_{j \notin D} \mathbf{b}_{max} \bar{P}_{i,j}^{(\bar{B})} = \mathbf{b}_{max} \cdot \sum_{j \notin D} \bar{P}_{i,j}^{(\bar{B})} = \mathbf{b}_{max} \cdot 1 = \mathbf{b}_{max} \qquad (16)$$

where $D$ is an arbitrary set of indices of deleted tokens. Note that whichever token set is deleted, the (renormalised) attention weights $\bar{P}_{i,j}^{(\bar{B})}$ always sum to 1.

The same follows analogously for $\mathbf{b}_{min}$ as elementwise lower bound on $\bar{\mathcal{B}}_i$.

**Bounds on $\bar{\mathbf{v}}_2$:**  Recall from Equation 6 that

$$\mathbf{v}_2 = \sum_j G([\mathcal{A}_j; \mathbf{b}_j]) = \sum_j \mathbf{g}_j \in \mathbb{R}^{d'} \qquad (17)$$

where $\mathbf{g}_j = G([\mathcal{A}_j; \mathbf{b}_j])$ for notational convenience. The function $G$ is a dense feed-forward neural network with softplus nonlinearity; consequently all values in $\mathbf{g}_j$ are strictly positive. Since each $\mathbf{g}_j$ has positive values, their sum will monotonically decrease if summands are removed, and monotonically increase as summands are added.

We consider two extreme cases of deleting word combinations: i) removing all but one word ii) removing precisely one word. These will be used to bound $\bar{\mathbf{v}}_2$ for any other number of deleted words which lie in between these extremes.

For the case that all but one words are removed (at position $r$), $\bar{\mathbf{v}}_2 = \mathbf{g}_r$, and the smallest values this expression can assume (elementwise) is $\min_{r=1,\dots,J}\{\mathbf{g}_r\}$. This is thus a lower bound on $\bar{\mathbf{v}}_2$ for sequences with only one word, and, due to the monotonicity of $\bar{\mathbf{v}}_2$ in its number of summands, also for $\bar{\mathbf{v}}_2$ under any combination of deleted words.

For the case of deleting only a single word at position $r$, one single summand is subtracted from $\mathbf{v}_2$: $\bar{\mathbf{v}}_2 = \mathbf{v}_2 - \mathbf{g}_r$, and this expression is bounded from above by $\mathbf{v}_2 - \min_{r=1,\dots,J}\{\mathbf{g}_r\}$. Again, due to the monotonicity of $\bar{\mathbf{v}}_2$ when deleting more symbols, this upper bound for single-word deletions is consequently also an upper bound for any combination of more deleted words.

To summarise, $\bar{\mathbf{v}}_2$ is bounded as follows:

$$\min_{r=1,\dots,J}\{\mathbf{g}_r\} \leq \bar{\mathbf{v}}_2 \leq \mathbf{v}_2 - \min_{r=1,\dots,J}\{\mathbf{g}_r\} \qquad (18)$$

