# OpenReview forum: "Towards Verified Robustness under Text Deletion Interventions"
_ICLR.cc/2020/Conference — Accept (Poster)_

### Official Review · AnonReviewer3 · 2019-10-25
**Official Blind Review #3**

**Rating:** 6

**Review:**

This paper proposes a model to verify the robustness of NLP models (change in the original probability), more specifically DAM, in the case of word removals in the input. The idea is given the lower and upper bound on the hidden state at previous layer, compute the new bound by propagating the bounding box around the hidden state at previous layer. The upper bound at the final layer is then compared with the label probability of the original input to assess if the probability increases or not. By training model with a hinge loss based on this verification method, they show that the model becomes more robust to word removals.

Overall, the paper is well written and the idea of using IBP with an attentive model seems to work empirically for SNLI datasets. But, the technical contribution feels incremental over previous approaches, especially Huang (2019). I have several questions related to some parts of the paper:

- Since upper and lower bounds are also propagated, do you backpropagate the gradients via these bounds or only via the original inputs?
- How sensitive is the label in SNLI dataset to word removal? For some label types, such as entailment, it might have less of an effect that for the others.
- How is the accuracy distributed wrt different label types?
- Since the accuracy of the proposed model drops the most, I am wondering how the verfied accuracy and accuracy are related during training? For example, can you show what is the verified accuracy with accuracy being close to the standard training?

**Experience Assessment:**

I have read many papers in this area.

**Review Assessment: Checking Correctness Of Derivations And Theory:**

N/A

**Review Assessment: Checking Correctness Of Experiments:**

I carefully checked the experiments.

**Review Assessment: Thoroughness In Paper Reading:**

N/A

---

> ### Author Response · Authors · 2019-11-13
> **Response to Review #3**
>
> Thank you for your review and comments.
>
> Comment:
> “[...] the technical contribution feels incremental over previous approaches, especially Huang (2019).”
>
> Response:
> Huang et al. (2019) indeed also made use of IBP, but address a different NLP task, and an over-sensitivity specification. Their work is adapted to a model without attention component, and in contrast to their work we focus on word deletion perturbations.
>
> Comment:
> “Since upper and lower bounds are also propagated, do you backpropagate the gradients via these bounds or only via the original inputs?”
>
> Response:
> Gradients are propagated backwards from the scalar training loss that is composed of both the standard log-likelihood objective, and the IBP contribution weighted by lambda. Note that the IBP contribution is a function of the bounds, which is a function of the model parameters, thus both loss contributions are a function of the model parameters. We can then compute gradients for how this loss changes depending on individual weights and biases, in the same vein as backpropagating errors from the standard differentiable training loss.
>
> Comment:
> “How sensitive is the label in SNLI dataset to word removal? For some label types, such as entailment, it might have less of an effect that for the others.”
>
> Response:
> All labels are affected, including the entailment label. In this context, Feng et al (2018) https://www.aclweb.org/anthology/D18-1407.pdf (Table 1) ran a study to measure human performance on reduced examples of the different classes, and the drop was sharpest for reduced samples of the entailment label (E).
>
> Comment:
> “How is the accuracy distributed wrt different label types?”
>
> Response:
> We found that verified accuracy is heavily skewed towards one label in SNLI: contradiction (58.2%), compared to neutral (1.3%) and entailment (0.0%, though nonzero). We furthermore observed a notable negative correlation (-0.19) of verified accuracy with lexical overlap of premise and hypothesis, even within the contradiction label, which has a lower rate of lexical overlap than the other two classes.
>
>
> Comment:
> “Since the accuracy of the proposed model drops the most, I am wondering how the verfied accuracy and accuracy are related during training? For example, can you show what is the verified accuracy with accuracy being close to the standard training?”
>
> Response:
> In our experiments we observed that verified accuracy and standard accuracy correlate negatively with one another, and prior work has shown that there is a tradeoff between robustness and standard accuracy (Tsipras et al. (2019), https://arxiv.org/abs/1805.12152). It actively hurts standard test accuracy when the model becomes verifiably less under-sensitive. This indicates that the signal that the model uses to form its prediction under standard training cannot be exploited (to the same extent) during verifiable training, and that the NLI task is then much harder to learn.

---

### Official Review · AnonReviewer2 · 2019-10-26
**Official Blind Review #2**

**Rating:** 8

**Review:**

This works considers the task of Natural Language Inference (NLI).
The question addressed is that SOTA NLI models tend to lead to
higher confidence when some parts are deleted from the "premise".
It is a problem known as under-sensitivity.
A method based on IBP is proposed to address this issue.
The idea of Interval Bound Propagation (IBP) is to use interval arithmetic to propagate
intervals and bound the variation of the target based
on variation of the input. In other words, one propagates
upper and lower interval bounds through the network.
The DAM model from (Parikh et al., 2016)
is studied in particular.

The paper is well written and easy to follow.

My only concern is about the relevance of approach based on DAM when
there are now more accurate models for this task. The paper is however
interesting and addressed a relevant topic.

Misc:
- transpose should be written with $^\top$ (not $^T$).



**Experience Assessment:**

I do not know much about this area.

**Review Assessment: Checking Correctness Of Derivations And Theory:**

N/A

**Review Assessment: Checking Correctness Of Experiments:**

I assessed the sensibility of the experiments.

**Review Assessment: Thoroughness In Paper Reading:**

I read the paper at least twice and used my best judgement in assessing the paper.

---

> ### Author Response · Authors · 2019-11-13
> **Response to Review #2**
>
> Thank you for your review and comments.
>
> Comment:
> “My only concern is about the relevance of approach based on DAM when
> there are now more accurate models for this task.“
>
> Response:
> The DAM model is indeed 5% behind the currently best model on SNLI, as measured in test accuracy (https://nlp.stanford.edu/projects/snli/). We would like to see verification extended to more sophisticated and deeper architectures, such as transformers, and we see the verification of an attention-based component, such as our work, as a stepping stone towards this goal.
>
> Comment:
> “transpose should be written with (not ).”
>
> Response:
> Thank you for your suggestion regarding notation, we have updated it throughout the paper.

---

### Official Review · AnonReviewer1 · 2019-10-29
**Official Blind Review #1**

**Rating:** 6

**Review:**

-- Overall --
This submission tackles to verify the “under-sensitivity” problem of neural network models in the natural language inference by ensuring modes do not become more confident in the predictions when arbitrary subsets of words from the input text are deleted. The authors developed new verification approaches based on decomposable attention mechanism with interval bound propagation (IBP), which can prove the under-sensitivity issue given a model and a particular sample. The experimental results on SNLI and MNLI show that the proposed approach leads to a much improved verified accuracy.

-- In general, “under-sensitivity” is a very critical problem for applying neural models in natural language understanding where powerful neural networks tend to capture spurious correlations from the biased datasets. This submission formulates “under-sensitivity” as a mathematical specification and then try to verify it with IBP verification. Although the used technique IBP is not new, it would interesting to have the verification in NLI models.

-- Section 5 is a bit unclear how to compute the IBP for deleting several words, and what is the output. It would be better to have a clear example for how this was computed.

-- As the author mentioned, the verification of under-sensitivity can also be done by using beam-search, although it is costly and not accurate. IBP is another more efficient option, but not the optimal neigher. Maybe consider to change the title as “efficient verification”?

-- Specific Questions --
The entire paper builds on decomposable attention. Is the same approach also applicable to other model types, or only single layer attention-based models?
Also, how this methods work for other NLI or NLU tasks?
In experiments, how the data augumentation penalize the model with a loss for specification violation? What does the equation look like?
Can you explain a bit more for IBP-training? How that hinge loss applies to the objective function? Is the IBP training differentiable?


**Experience Assessment:**

I have read many papers in this area.

**Review Assessment: Checking Correctness Of Derivations And Theory:**

I assessed the sensibility of the derivations and theory.

**Review Assessment: Checking Correctness Of Experiments:**

I assessed the sensibility of the experiments.

**Review Assessment: Thoroughness In Paper Reading:**

I read the paper at least twice and used my best judgement in assessing the paper.

---

> ### Author Response · Authors · 2019-11-13
> **Response to Review #1**
>
> Thank you for your comments and feedback.
>
> Comment:
> “Section 5 is a bit unclear how to compute the IBP for deleting several words, and what is the output. It would be better to have a clear example for how this was computed.”
>
> Response:
> Thank you for this suggestion, we have included a corresponding section into the appendix that expands on this.
>
> Comment:
> “Is the same approach also applicable to other model types, or only single layer attention-based models?“
>
> Response:
> Interval Bound Propagation as verification technique can be applied to other types of network architectures as well, such as CNN or feed-forward components (we refer to prior work on this in the Related Work section). We focused on the DAM architecture, where bound propagation has to be adapted to an attention-component, and believe that several aspects of this can be transferred to transformer-based architectures (e.g. attention renormalisation). There is however a caveat: deeper models tend to produce looser IBP bounds, and we would consequently expect lower verification rates for deep transformer models.
>
> Comment:
> “Also, how this methods work for other NLI or NLU tasks?”
>
> Response:
> Under-sensitivity has been established for NLI (with transferable attacks between models), and also for SQuAD (Feng et al. (2018)). IBP has been used with other neural layer types, and it should in a similar vein be possible to derive bounds for models used on other NLU tasks. It would be interesting to see how verification of under-sensitivity specifications would play out elsewhere, and we hope that future research will address this problem.
>
> Comment:
> “In experiments, how the data augumentation penalize the model with a loss for specification violation? What does the equation look like?”
>
> Response:
> In data augmentation training we compute, for a standard data point x, a randomly perturbed data point x’ that has a random subset of words removed. We derive a hinge loss contribution from this as follows: lambda * max{0, P(y|x’) - P(y|x)}, which is added to the standard training loss, where lambda is a scalar hyperparameter.
>
> Comment:
> “Can you explain a bit more for IBP-training? How that hinge loss applies to the objective function? Is the IBP training differentiable?”
>
> Response:
> There is an upper bound (IBP) for the model output corresponding to the gold label y, and a nominal probability P(y|x). Both depend on all model parameters, and the difference delta between the two is again a function of all model parameters. This difference delta is fed into a hinge function lambda * max{0, delta} and we add it to the standard training loss, again with a scalar hyperparameter lambda. The resulting loss is then differentiable, and during model training the parameters are tuned to minimise the loss via gradient-based optimisation.

---

### Official Review · AnonReviewer4 · 2019-11-02
**Official Blind Review #4**

**Rating:** 3

**Review:**

This work is an application of interval bound propagation on evaluating the robustness of NLI model. This work is well-motivated, assuming that the confidence of a neural model should be lower when part of the sentence is missed. However, the application of vanilla IBP is quite limited in certain model architectures. In this work, the author considers specifically the decomposable attention model, which is a very shallow network, and not a state-of-the-art model anymore. It is non-trivial to adapt the proposed method to other more advanced models, such as the ones based on the Transformer model. Hence, this work does not make enough contribution to be accepted.

**Experience Assessment:**

I do not know much about this area.

**Review Assessment: Checking Correctness Of Derivations And Theory:**

I assessed the sensibility of the derivations and theory.

**Review Assessment: Checking Correctness Of Experiments:**

I assessed the sensibility of the experiments.

**Review Assessment: Thoroughness In Paper Reading:**

I read the paper at least twice and used my best judgement in assessing the paper.

---

> ### Author Response · Authors · 2019-11-13
> **Our Model Choice, Context of this Work, and Concrete Contributions.**
>
> Thank you for taking the time to review our work and for your comments.
>
> Comment:
>  “author considers […] decomposable attention model, which is a very shallow network, and not a state-of-the-art model anymore”
>
> Response:
> The focus of our paper is to make neural models conform to the under-sensitivity specification. The DAM architecture was chosen as it is one of the models for which under-sensitivity has originally been established (see Feng et al. (2018) https://www.aclweb.org/anthology/D18-1407.pdf, Section 5.2). You are right to observe that the DAM model we chose to verify is 5% behind the currently best model on SNLI, as measured in test accuracy (https://nlp.stanford.edu/projects/snli/).
>
> Comment:
> “the application of vanilla IBP is quite limited in certain model architectures. [...]non-trivial to adapt the proposed method to other more advanced models, such as the ones based on the Transformer model.”
>
> Response:
> Neural network verification is a very challenging problem to address, and an active research area in the community. In the image domain IBP can only scale to at most 10 layers on CIFAR10, and not to ImageNet or very deep networks (Gowal et al. (2018), https://arxiv.org/pdf/1810.12715.pdf). Specification verification in NLP is itself a relatively new topic, and we expect that straightforward application of IBP to a deep transformer model would lead to interval bounds quickly growing with network depth, an obstacle that will have to be addressed to certify deeper networks.
>
> Comment:
> “Hence, this work does not make enough contribution to be accepted”
>
> Response:
> We agree that certification of other model architectures, such as the transformer, is a challenging and worthy goal. We see our work and model choice (smaller, and with transferable architectural components, such as attention) as a step in this direction. We believe that our contributions on:
> 1) Evaluating and treating undersensitivity
> 2) Use of bound propagation principles to verify attention-based layers
> 3) Training of models that are verifiably not undersensitive
> are all useful steps towards understanding and extending the robustness of NLP models.

---

### Decision · Program_Chairs · 2019-12-19

**Decision:**

Accept (Poster)

**Comment:**

This paper deals with the under-sensitivity problem in natural language inference tasks.  An interval bound propagation (IBP) approach is applied to predict the confidence of the model when a subsets of words from the input text are deleted.  The paper is well written and easy to follow.  The authors give detailed rebuttal and 3 of the 4 reviewers lean to accept the paper.